# Agreement between patient's description of abdominal symptoms of possible upper gastrointestinal cancer and general practitioner consultation notes: a qualitative analysis of video-recorded UK primary care consultation data

Victoria Hardy ,[1] Juliet Usher-Smith ,[1] Stephanie Archer ,[1] Rebecca Barnes,[2] John Lancaster,[1] Margaret Johnson,[1] Matthew Thompson,[3] Jon Emery ,[4] Hardeep Singh ,[5,6] Fiona M Walter ,[1,7]

For numbered affiliations see end of article.

**Correspondence to**
Victoria Hardy;
veh29@medschl.cam.ac.uk

## ABSTRACT

**Introduction** Abdominal symptoms are common in primary care but infrequently might be due to an upper gastrointestinal (UGI) cancer. Patients' descriptions may differ from medical terminology used by general practitioners (GPs). This may affect how information about abdominal symptoms possibly due to an UGI cancer are documented, creating potential missed opportunities for timely investigation.

**Objectives** To explore how abdominal symptoms are communicated during primary care consultations, and identify characteristics of patients' descriptions that underpin variation in the accuracy and completeness with which they are documented in medical records.

**Methods and analysis** Primary care consultation video recordings, transcripts and medical records from an existing dataset were screened for adults reporting abdominal symptoms. We conducted a qualitative content analysis to capture alignments (medical record entries matching patient verbal and non-verbal descriptions) and misalignments (symptom information omitted or differing from patient descriptions). Categories were informed by the Calgary-Cambridge guide's 'gathering information' domains and patterns in descriptions explored.

**Results** Our sample included 28 consultations (28 patients with 18 GPs): 10 categories of different clinical features of abdominal symptoms were discussed. The information GPs documented about these features commonly did not match what patients described, with misalignments more common than alignments (67 vs 43 instances, respectively). Misalignments often featured patients using vague descriptors, figurative speech, lengthy explanations and broad hand gestures. Alignments were characterised by patients using well-defined terms, succinct descriptions and precise gestures for symptoms with an exact location. Abdominal sensations reported as 'pain' were almost always documented compared with expressions of 'discomfort'.

## STRENGTHS AND LIMITATIONS OF THIS STUDY

⇒ This is the first study to explore similarities and differences between how patients describe and general practitioners (GPs) document abdominal symptoms that could be due to an upper gastrointestinal cancer using naturalistic UK primary care consultation data.

⇒ We did not have access to patients' previous consultation history and so could not determine the clinical importance of symptom information that was either not documented in medical records, or did not reflect the patient's history.

⇒ As our sample was mostly white British and native English speakers, patterns in how patients described and GPs documented abdominal symptoms may not be transferable to practices serving patient populations with different characteristics.

**Conclusions** Abdominal symptoms that are well defined or communicated as 'pain' may be more salient to GPs than those expressed vaguely or as 'discomfort'. Variable documentation of abdominal symptoms in medical records may have implications for the development of clinical decision support systems and decisions to investigate possible UGI cancer.

## INTRODUCTION

Sir William Osler, lauded as the father of modern medicine, implored his medical students to 'listen to your patient; he is telling you the diagnosis'.[1] Over a century later, this adage remains germane. As depicted in the National Academy of Medicine's conceptual model of the diagnostic process, an accurate and timely diagnosis is predicated

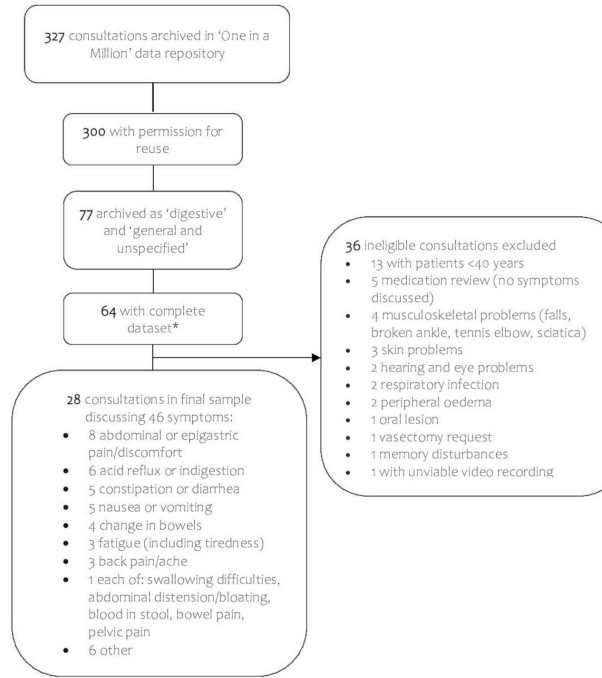

327 consultations archived in 'One in a Million' data repository

↓

300 with permission for reuse

↓

77 archived as 'digestive' and 'general and unspecified'

↓

64 with complete dataset*

↓

28 consultations in final sample discussing 46 symptoms:
- 8 abdominal or epigastric pain/discomfort
- 6 acid reflux or indigestion
- 5 constipation or diarrhea
- 5 nausea or vomiting
- 4 change in bowels
- 3 fatigue (including tiredness)
- 3 back pain/ache
- 1 each of: swallowing difficulties, abdominal distension/bloating, blood in stool, bowel pain, pelvic pain
- 6 other

36 ineligible consultations excluded
- 13 with patients <40 years
- 5 medication review (no symptoms discussed)
- 4 musculoskeletal problems (falls, broken ankle, tennis elbow, sciatica)
- 3 skin problems
- 2 hearing and eye problems
- 2 respiratory infection
- 2 peripheral oedema
- 1 oral lesion
- 1 vasectomy request
- 1 memory disturbances
- 1 with unviable video recording

*Video recording, transcript, and medical record entry

**Figure 1** Flow diagram of included consultations.

on clinicians collecting sufficient data from the patient history.[2] Communication skills teaching is integral to medical training,[3] but suboptimal information gathering is common during the medical interview[4–6] and has been implicated in diagnostic error in primary care settings.[5]

Abdominal symptoms are common in patients presenting to primary care.[7] While such symptoms far more frequently herald benign conditions,[8] they are also described by patients diagnosed with upper gastro-intestinal (UGI) cancers (ie, gastric, oesophageal and pancreatic) as well as other cancers within the abdomen.[7] Compared with 'alarm' symptoms which have relatively high positive predictive value (PPV), abdominal symptoms alone do not meet the 3% PPV threshold recommended by the National Institute for Health and Care Excellence (NICE) NG12 (2015) for a 'fast-track' suspected cancer referral.[9] Compounded by a lack of screening programmes or biomarkers reliably raising suspicion of UGI cancer in low prevalence populations,[10] patients with UGI cancers are disproportionately burdened with lengthy diagnostic intervals and late-stage diagnosis which are associated with worse survival and patient experiences of healthcare.[11] While the novel 'CytoSponge' test (for Barrett's oesophagus, a precursor lesion for oesophageal cancer[12]) is a promising development for early detection, effective triage of abdominal symptoms for a timely UGI cancer diagnosis, remains currently largely contingent on the patient history.

Symptoms constitute subjective changes in bodily sensations that reflect a perceived departure from the body's 'normal' state.[13 14] Influenced by sociocultural[13] and experiential factors,[14] patients' internal representation

of such bodily changes may be communicated differently from standardised medical terminology used by general practitioners (GPs).[15] Studies of UGI cancer patients' symptom appraisal have shown that patients use a wide range of words and phrases to conceptualise abdominal changes[16 17] that are inconsistent with medical definitions familiar to GPs.[17 18] Without a common language, potentially significant bodily sensations risk being overlooked or misinterpreted and the clinical picture distorted as 'lay' descriptions are translated into equivalent medical nomenclature.[15] Consequently, what GPs document in medical records may not be a complete or accurate representation of the patient's history.[19]

As well as being important at the patient level for subsequent management decisions, the ways GPs (and other equivalent primary care providers outside the UK) document patients' bodily sensations in medical records is important for population approaches to early cancer detection. In the UK, coded symptom data is harvested from primary care medical records to derive symptom-based guidelines and prediction tools that prompt GPs to consider certain investigations for patients' reaching the 3% PPV threshold.[20–23] There is also mounting interest in harnessing Natural Language Processing techniques to incorporate free-text symptoms.[24 25] The utility of symptom-based tools for timely investigation of UGI (and other) cancers is contingent upon symptoms in medical records mirroring those described by patients during primary care consultations. However, the extent to which abdominal symptoms documented by GPs reflects the patient's history remains unclear.

To address this research need, we compared video-recorded primary care consultations featuring new or ongoing abdominal sensations that could be due to an UGI cancer with medical record documentation and explored similarities and differences between patient descriptions of bodily sensations and the way GPs document these as symptoms. We aimed to explore how abdominal symptoms are recorded in the medical record to identify the characteristics of patients' descriptions underpinning GP variation in the medical record entries. We also consider the implications of this on opportunities for earlier diagnosis of UGI cancers. Although other healthcare practitioners working at an advanced level of practice in primary care play an important role in the early recognition of symptoms of possible cancer, this study builds on existing observational research that has examined GP's appraisal[26] and documentation of patient's symptoms.[27]

## METHODS
### Design
A cross-sectional descriptive study of video-recorded primary care consultations with linked medical record data entries from the 'One in a Million' primary care consultation archive, analysed using qualitative content analysis.

**Table 1** Sample characteristics

| Patient characteristics | No of patients (N=28) | GP characteristics | No of GPs (N=18) |
|---|---|---|---|
| Sex | | Sex | |
| Male | 12 | Male | 10 |
| Female | 16 | Female | 8 |
| Ethnicity* | | Ethnicity | |
| White British/white other | 26 | White British/white other | 18 |
| Mixed/multiple ethnic groups | 1 | Mixed/multiple ethnic groups | – |
| Age | | Age | |
| 30–39 | NA | 30–39 | 5 |
| 40–49 | 5 | 40–49 | 5 |
| 50–59 | 6 | 50–59 | 7 |
| 60–69 | 6 | 60–69 | 1 |
| 70–79 | 9 | 70–79 | – |
| 80–89 | 2 | 80–89 | – |
| Main language | | Main language | |
| English | 27 | English | 18 |
| Country of birth | | Country of birth | |
| UK | 28 | UK | 18 |
| Highest educational attainment | | Years since qualified as a GP | |
| Degree or higher degree | 8 | 1–10 | 5 |
| A/AS levels or advanced diploma | 2 | 11–20 | 6 |
| Professional or vocational qualifications | 6 | 21–30 | 6 |
| Apprenticeship | 3 | 31+ | 1 |
| No qualifications | 4 | | |
| IMD quintiles | | | |
| First (least deprived) | 6 | | |
| Second | 8 | | |
| Third | 4 | | |
| Fourth | 5 | | |
| Fifth (most deprived) | 5 | | |

*Not reported (n=1).
GP, general practitioner; IMD, Index of Multiple Deprivation; NA, not applicable.

## Data source

The 'One in a Million' dataset was derived from the 'Bristol Archive project,' a prospective observational cohort study that sought to create a repository of routine primary care consultation data available for use in future research.[28] Between July 2014 and April 2015, 12 purposively sampled general practices in the West of England recorded unselected consultations with adult patients (aged 18 years and above) who presented to primary care over two to three half days.[29] A total of 327 consultations were recorded with linked patient, GP and practice demographic data, and available medical record entries. 89.8% (n=300) of patients consented to selected datasets being reused in future research, subject to specific ethical approval. All these recordings were anonymised for spoken identifiers, transcribed verbatim and coded for

their content according to 16 'problem type(s)' based on the International Classification of Primary Care-second edition classification scheme.[29] Consultations were assigned more than one code if multiple problems were discussed. We requested all available data for 84 consultations archived as 'digestive' or 'general and unspecified'[29] to yield data directly relevant for addressing our research question.

## Eligibility criteria and assessment

We included consultations if there was: (A) an accompanying videorecording, transcript and medical record entry, and (B) communication of any new or ongoing abdominal and systemic sensations that could reflect an underlying UGI cancer. We limited inclusion to patients aged 40 years and above, because UGI cancers below this

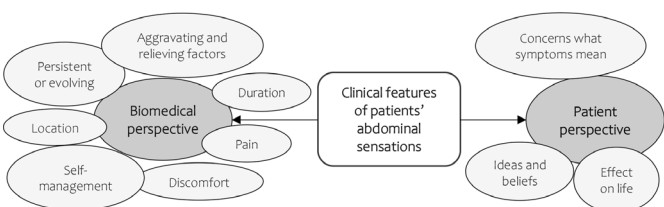

**Figure 2** Final categories representing the clinical features of abdominal sensations discussed in primary care consultations based on the 'gathering information' domains of the Calgary-Cambridge model.

age are rare.[30] Consultations of patients known to have a previous or current cancer diagnosis based on information disclosed during the video recording were excluded, as a history of cancer provokes more suspicion of serious illness among GPs and patients, and we were interested in GPs' appraisal of patient's bodily sensations in an as-yet undiagnosed population. We also excluded consultations during which no relevant bodily sensations were communicated. This was determined by abstracting all bodily sensations discussed in consultations into a preprepared Microsoft Excel form and assessing each as 'include', 'exclude' or 'unsure'. Eligibility assessment was initially undertaken by VH (a non-clinical PhD student) before final adjudication of relevant bodily sensations by FMW (a senior academic GP). Consultations for which there were no relevant bodily sensations were excluded at this juncture.

### Sampling
As this study was an analysis of a pre-existing archived dataset and no further data collection was possible, we included all consultations of symptomatic patients that met study eligibility criteria.

### Data analysis
Data were analysed using a manifest qualitative content analysis, following the four phases outlined by Erlingsson *et al*: 'familiarising oneself with the data', 'dividing and condensing the text into meaning units', 'formulating codes' and 'developing categories and themes'.[31] A manifest qualitative content analysis was selected as it is the most suitable analytical technique for exploring and describing visible and explicit patterns (as opposed to the underlying meaning of text as in latent content analysis) across data sources that incorporate visual and textual communication data.[32–34] Data analysis was conducted in Excel by VH, with analysis facilitated by SA (senior non-clinical researcher), and data interpretation verified by FMW and JU-S (senior academic GPs) and PPI (JL and MJ).

We employed a mixed deductive and inductive approach whereby the analytical framework was informed by the 'gathering information' domains of the Calgary-Cambridge guide. We selected the Calgary-Cambridge guide over other possible instruments (eg, the Roter Interaction Analysis System) as it is an evidence-based teaching and assessment resource that is used in medical training programmes internationally to develop doctors' verbal and non-verbal communication skills and structure consultations. The guide delineates the consultation into distinct phases, including 'initiating the session', 'gathering information', 'physical examination', 'explanation and planning' and 'closing the session'. The 'gathering information' component guides doctors through the collection of symptom information relating to the biomedical perspective, patient perspective and background information that should be elicited during the patient history. Thus, the Calgary-Cambridge guide offers an ecologically valid framework for guiding the deconstruction and analysis of the information communicated and documented in primary care consultations.

Analysis begun by cross-referencing transcripts with video recordings to verify accuracy. Transcripts, video recordings and medical record entries were then reviewed repeatedly, individually and concomitantly, to become familiar with the content. Video recordings were reviewed alongside transcripts to observe the nonverbal body language exhibited by patients and GPs. Six nonverbal aspects of communication were annotated in parentheses adjacent to the relevant text. These included posture, proximity, touch, body movements, facial expression and eye behaviour.[35]

Short verbatim excerpts or 'meaning units' (encompassing nonverbal gestures) were juxtaposed with medical record entries using an adapted consultation proforma originally developed for examining the content of different primary care data sources.[36] Throughout analysis, excerpts from each consultation were kept together to allow contextualisation of the data. 'Meaning units' were condensed by VH into an abbreviated narrative capturing the essence of patient's reason for presentation. Care was taken to preserve the patient voice by weaving verbatim descriptions from excerpts into a reconfigured narrative. Patient reported bodily sensations were then compared with symptoms documented by GPs in the patients' medical record, and alignments and misalignments in documentation coded. Alignments were symptom information in medical records that both mirrored the content and verbal and non-verbal descriptors patients communicated; misalignments were symptoms that had either not been documented or differed from information given by patients. This task was independently undertaken by VH and two patient and public representatives (PPI) (MJ and JL); differences were resolved through further discussion and consensus with FMW and JU-S. Coded excerpts constituted the unit of analysis and encompassed multiple features of an individual symptom.

Coded alignments and misalignments were separated from original data sources and, with regular input from study authors, iteratively grouped into descriptive categories, informed by the symptom features listed in the 'gathering information' domains.[35]

**Table 2** Differences in the characteristics of patients' verbal and non-verbal descriptions of abdominal sensations for alignments and misalignments

| Clinical feature | Alignments | Misalignments |
|---|---|---|
| **Domain: Biomedical perspective** | | |
| Persistent or evolving | Well-defined terms, articulating changes in quality, frequency, and intensity of worsening sensations | Non-definitive, figurative speech for bodily sensations that were worsening, improving but not resolved, or re-appraised as worsening |
| Location of pain/ discomfort | Well-defined or medical terms, succinctly communicated; non-verbal gestures pointing to site of pain/discomfort | Difficulty articulating precise site of pain/ discomfort; broad or global non-verbal gestures not indicating site of pain/discomfort |
| Aggravating and relieving factors | Communication of a relationship between activities of daily living and onset or worsening of sensations | Infrequent provocation of sensations; lengthy explanations regarding activities that provoked or worsened sensations; activities that relieved sensations |
| Discomfort | Adjectives or figurative speech for unpleasant sensations | Adjectives or figurative speech for unpleasant sensations |
| Pain | Pain explicitly articulated with qualifying adjectives, similes and verbs describing an aversive sensation | Pain explicitly articulated with qualifying adjectives, similes and verbs describing an aversive sensation |
| Duration | Communicating the day or number of weeks or months since onset of symptoms | Vague or oblique sentences without offering a timeframe |
| Self- management | Reporting sensations that were controlled or settling with medication | Reporting suboptimal therapeutic effect of medication |
| **Domain: Patient's perspective** | | |
| Effects on life | (No differences in patient descriptions between alignments and misalignments) | |
| Ideas and beliefs | Plausible and convincing explanation as to underlying cause related to medication side effects, flare-up of pre-existing condition, diet, or stress | Lack of a justified explanation for possible cause of sensations |
| Concerns about what symptoms might mean | Explicit concerns voiced about the possibility of cancer or more broad concerns that something was not right | Verbalising concerns about the possibility symptoms had a sinister cause, but not explicitly mentioning cancer |

Final categories were recontextualised, and the features of patient's verbal descriptions and non-verbal gestures underpinning alignments and misalignments explored. Attention was given to isolated and discrepant excerpts to illuminate as much diversity of symptom expression encountered in general practice. We also explored differences in GP documentation by GP and patient demographic characteristics.

### Patient and public involvement

Two patient and public involvement representatives (JL and MJ) actively advised the research group from study inception to manuscript preparation. Both representatives were involved in the analysis and interpretation of the data, and will facilitate dissemination of study findings.

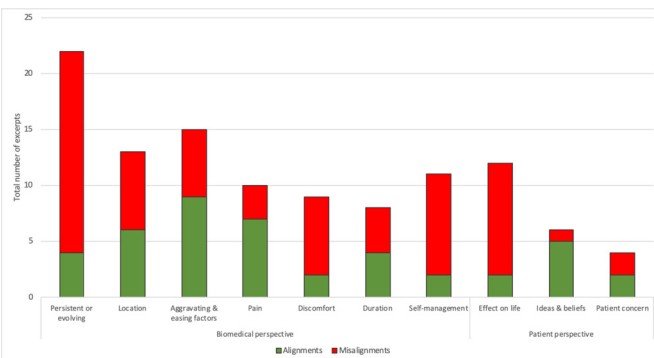

**Figure 3** Frequency of alignments and misalignments for each feature of patients' abdominal and systemic bodily sensations, according to the 'gathering information' domain of the Calgary-Cambridge guide.

### RESULTS

### Included consultations

Of the 77 consultations requested from the 'One in a Million' data repository, 64 were accompanied with a complete dataset, of which 28 met study eligibility criteria and were included in the final analysis (figure 1). Participants comprised 28 patients, 1 caregiver and 18 GPs. Overall, patients presented with 46 new or ongoing problems, captured in a total of 110 textual excerpts that were relevant for this analysis.

## Sample characteristics

Characteristics of included patients and GPs are presented in table 1 (the one caregiver's characteristics were unavailable). Patients were mostly female (n=16), aged 40–89 years, with varying educational attainment. GPs were aged 30–69 years and had been qualified as a GP for up to 31 years. Patients and GPs were all born in the UK and spoke English as their primary language. Apart from one patient of mixed ethnicity, our sample was white British. Patients presented with a variety of abdominal and systemic bodily sensations (online supplemental file A).

## GP alignments and misalignments

Overall, there was more evidence of GP misalignments (n=67) than alignments (n=43), with a median of 2 (range: 0–8) misalignments and 1 (range: 0–5) alignments, per consultation.

We identified 10 categories reflecting 10 clinical features which were informed by the 'gathering information' domains of the Calgary-Cambridge guide (figure 2 and table 2).[35] Of these 10 categories/features, 7 related to the 'biomedical perspective' and 3 to the 'patient's perspective' (figure 2). Most misalignments occurred for abdominal sensations that were 'persistent or evolving' or expressed as 'discomfort', or when documenting the 'location', 'effect on patients' life' and 'self-management' of abdominal sensations (figure 3). Alignments were proportionally most common for 'aggravating and easing' factors, when abdominal sensations were described as 'pain', and when capturing patient's ideas and beliefs. Findings for all 10 clinical features are detailed below. A summary of the main differences in patient descriptions of each feature between alignments and misalignments is presented in table 2.

## Domain: biomedical perspective

### Persistent or evolving

Abdominal sensations that were persistent or evolving in frequency or intensity were described by patients most often (on 22 occasions), though were poorly reflected in medical records. Alignments occurred when patients used well-defined or concrete terms to indicate that their bodily sensations were progressing:

| | |
|---|---|
| **Patient:** *'my poo has got a lot freer and a lot looser probably in about the last six weeks. Most of the time when I go to the loo I will do some sort of poo as well at the same time. (shakes head)'* (P22: F, 40–49 years) | **Medical records:** last *6 weeks loose stools and more freq (GP19)* |

Abdominal sensations described in non-definitive ('not getting better') or figurative speech, that were receding but not resolved, or that patients reappraised as worsening, were characteristic of misalignments. In these instances, bodily sensations were erroneously documented as normal, resolving or stabilised, respectively:

| | |
|---|---|
| **Patient:** ;*bowels are a bit sticky (intermittent eye contact, squeezing toothpaste motion with both hands) but I've been eating prunes and apricots and (laughs)*<br>**GP:** ;*And do you tend to have problems in that area anyway?;*<br>**Patient:** *'Not normally (looks upwards) and that was part of the symptoms last week'* (P1: F, 60–69) | **Medical records:** *constipation improving with prunes. (GP23)* |

### Location of pain/discomfort

The location of patients' problem was discussed in relation to abdominal 'pain' or 'discomfort' on 13 occasions. Misalignments were slightly more common than alignments. They occurred when pain/discomfort was communicated to be diffuse, move around, radiate to another location, or in instances when patients were unable to articulate the precise location and relied on large non-verbal hand gestures.

| | |
|---|---|
| **GP:** *'So where are you point- (tilts head) is it the middle of the belly, upper belly?'*<br>**Patient:** *''m not really sure, (looking down at abdomen) but I'm just going like that with it, (rocking back and forth both arms cradling abdomen) because it…'* (P18: F, 50–59 years) | **Medical records:** *nothing recorded in relation to location (GP15)* |

Location of abdominal pain/discomfort was accurately documented when a specific anatomical site was isolated, communicated in succinct and well-defined (sometimes medical) terms, and supplemented with non-verbal gestures that reinforced the specific site of their pain/discomfort:

| | |
|---|---|
| **Patient:** *'I've had an abdominal discomfort for about four or five weeks. It's just sore, really, just below the sternum. (tips of fingers press into epigastric region).* (P28: F, 50–59 years) | **Medical records:** *epigastric pain (GP12)* |

### Aggravating and relieving factors

Aggravating and relieving factors referred to activities of daily living that provoked, aggravated or worsened patients' bodily sensations. GPs' medical record entries accurately reflected information patients communicated in relation to this feature, when a causal relationship

between an activity (ie, stress, medication and certain foods) and the onset or worsening of sensations was articulated:

| Patient: *'It's like cucumber, red onion, tomatoes; it's certain things. I love to eat jacket potatoes, it just sets off heartburn. (P19: F, 40–49 years)* | Medical records: *Gastro-oesophageal reflux: with certain foods (GP18)* |

Information about activities aggravating bodily sensations were omitted from medical records when they were described as infrequent or improving, gave lengthy explanations of the precipitating factors, or in one instance, communicated an activity that eased bodily sensations:

| Patient: *'I have weeks now as long as I don't eat sugar and I don't have any stress it's often really quite good(…)At the moment it's a bit worse because I'm fighting for my son's funding and that gets me stressed. (shifts in chair, strokes forehead) Mostly it's so much better.' (P25: F, 50–59 years)* | Medical records: *nothing recorded in relation to aggravating and relieving factors (GP20)* |

### Discomfort

Though there were no differences in patients' communication of discomfort between alignments and misalignments, patients used a range of adjectives and similes to describe abdominal sensations that were causing discomfort. Subtle verbal descriptors, such as 'a nuisance' or as though 'something was pressing down' conveyed discomfort as a sensation that was interfering and unpleasant, and distinct from 'pain'. Patient's descriptions of discomfort were omitted or inaccurately reflected by GPs in most instances:

| Patient: *'I can still feel a little something there (left hand touches hollow of neck)(…) a sensation up here but it can't be a physical blockage because they looked(…)it's just I feel like you've got an airlock…" (P20: F, 70–79 years)* | Medical records: *nothing recorded in relation to the quality of the sensation (GP18)* |

### Pain

GPs tended to document 'pain' when it was articulated. There were no observable differences in how patients described pain between alignments and misalignments. However, patients typically qualified 'pain' with supplementary adjectives, verbs and similes, including 'terrible', 'stabbing' and 'like waves, like contractions' that communicated an aversive sensation. Such qualifiers were characteristic of both alignments and misalignments and were rarely captured in medical records.

### Duration

The duration of patients' bodily sensations was infrequently described. GPs' accurately documented what patients communicated in relation to this feature in half of the eight instances it was discussed. Offering a discrete timeframe by giving the day of the week or number of weeks or months that had elapsed since the onset of bodily sensations, was equally characteristic of misalignments and alignments:

| GP: *'So how long have you been poorly, now?'* Patient: *'Well, it wasn't last week. What's today? Tuesday. It was the Friday week. Last Friday week, yes' (nods head) (P3: F, 70–79 years)* | Medical records: *lower abdo pelvic pain for 10 days (GP5)* |

In contrast, vague or open-ended descriptions of the time since onset were slightly more typical of this feature being omitted from medical records.

### Self-management

Patients described use of prescription and over-the-counter medication taken to manage their bodily sensations. GP misalignments, which were more commonplace than alignments, were mostly characterised by lengthy explanations that the medication had eased but not resolved, or had failed to appreciably control the severity of bodily sensations:

| Patient: *'I took those capsules. I took one in the morning before anything else. And I took one at night before a meal(…)It [upper abdominal pain] settled a little bit. But it's still there (right hand presses upper abdomen) (P17: F, 70–79 years)* | Medical records: *nothing recorded in relation to self-management (GP14)* |

In the two instances GP's documentation matched what patients had communicated, patients asserted that the medication was either helping to settle or control bodily sensations, or was having no effect:

| Patient: *'I started taking prednisolone…(for colitis flare-up). Well, it is working. (eye contact, nods head) I've been taking them for a couple of weeks and the condition is improved…(P21: M, 70–79 years)* | Medical records: *ulcerative colitissettled now with use of oral steroids (GP19)* |

### Domain: the patient's perspective
### Effect on life

The effect of patients' bodily sensations on functional and social aspects of their life was discussed on 12 occasions but was rarely documented in medical records. There were no clear language differences between misalignments and alignments.

### Ideas and beliefs

Patient's ideas and beliefs were documented on most occasions, typically when patients offered a plausible explanation as to what may be causing their bodily sensations. Patients attributed sensations to a variety of factors including medication side effects, flare-up of a pre-existing condition, diet and stress, which on occasion matched GPs ideas and beliefs. In the one instance that patient's ideas and beliefs were omitted from records, the patient was unable to justify the possible cause.

### Concern about what symptoms might mean

Patients rarely expressed concerns about the cause or seriousness of their bodily sensations. In the two instances where patient's concerns were documented in medical records, patients had either relayed their partner's concern about 'looking drawn' or explicitly asked the GP whether their bodily sensations could be due to an underlying cancer:

| | |
|---|---|
| **Patient:** '*You don't think it's cancerous basically? (shakes head, right hand holding shirt collar)*' (P25: F, 50–59 years) | **Medical records:** *Very worried about cancer* (GP20) |

Patients' concerns were typically not documented when the possibility of a sinister cause was voiced but cancer was not directly articulated:

| | |
|---|---|
| **Patient:** '*There's nothing bad really down there? (touches upper abdomen with both hands)(...) Because it swells up quite a bit when it happened*' (P17: F, 70–79 years) | **Medical records:** *nothing recorded in relation to patient concerns* (GP14) |

### GP and patient characteristics

The frequency of alignments and misalignments for each clinical feature was insufficient for stratifying data to identify GP characteristics (sex, age, years since qualified as a GP) or patient characteristics (sex, age, educational attainment, social deprivation level) as additional sources of GP variation in how abdominal symptoms were documented.

## DISCUSSION
### Summary of key findings

A wide range of clinical features pertaining to abdominal and systemic sensations were discussed between patients and GPs during primary care consultations. There was a gap between what patients communicated and the information about these bodily sensations GPs documented in medical records. Vague descriptors, figurative speech and lengthy explanations of bodily sensations without an exact location, characterised features of patients' history that was not recorded, documented incompletely, or did not match what patients verbalised. Clinical features that were well-defined and succinct with precise gesticulations for abdominal sensations were well documented. Abdominal 'pain' was almost always documented compared with 'discomfort'. 'Discomfort' was conveyed using descriptors that indicated sensations that were unpleasant, whereas 'pain' was communicated as an aversive sensation. Abdominal features were omitted from medical records when large hand gesticulations, which supplemented or superseded verbal descriptions for the location of 'pain' or 'discomfort', were used.

### Strengths and limitations

A strength of this work is that our findings are derived from direct comparison of naturalistic data, verified through video observation and medical records. To our knowledge, this is the only study to explore how patients describe undifferentiated symptoms of possible cancer during a consultation and compare this to what is subsequently documented in patients' medical record. Patients in this study communicated a range of abdominal symptoms that patients with UGI cancer present with before diagnosis. This affords credible insight into the different ways patients may attempt to convey abdominal symptoms when presenting to primary care. By structuring our analysis around the 'gathering information' domains of an established communication teaching tool,[37] we were able to characterise nuances in patients' communication of abdominal sensations according to the clinical features doctors are trained to explore during the medical interview and as such are directly applicable to medical training programmes. By exploring how patients describe symptoms that *could* be caused by an UGI cancer prior to a suspected cancer referral, we were able to gain insight into the phenotypic presentation of lower-risk symptoms that often do not prompt timely investigation of possible UGI cancer (or other serious illness, including non-UGI cancers).

This study does have some limitations. Our findings reflect the immediately observable aspects of patient and GPs' verbal and non-verbal communication, rather than latent meaning. Thus, misalignments are not necessarily synonymous with a poor history as GPs may have accurately interpreted patients intended meaning despite literal differences in written documentation.[38] What GPs documented may have been affected by individual characteristics of patients and GPs, patients' agenda, and the structure and relative brevity of medical records in UK general practice, which were beyond the scope of this study.[39] A complete patient history is more than the sum of its parts; parsing abdominal sensations from the patient narrative (as in our approach) potentially negates correct interpretation of the overall clinical picture. As we did not have access to records of patients' prior medical history or final diagnosis, we could not determine whether symptoms documented (or omitted) were truly indicative of serious disease. We therefore identified that there was a difference between what patients communicated and what GPs documented, not whether this difference was clinically important. Our sample was almost exclusively White British and native English speakers so

our findings may not be transferable to practices serving patient populations with different characteristics. The focus of this study was limited to GPs. Therefore, our findings have most applicability to GPs in the UK and countries with similar healthcare structure. Our findings are likely to be relevant to other primary care providers in the UK though. For example, advanced nurse practitioners and physician associates who also see and manage patients with symptoms that could be caused by cancer. This study analyses the content of UK face-to-face consultations before COVID-19 prompted a shift to teleconsulting for triaging new patient problems in primary care.[40] Although face-to-face consultations are gradually resuming, remote consultations remain ubiquitous and may affect how patients communicate and what GPs document. Our findings are therefore unlikely to reflect communication or documentation practices occurring from remote patient–GP interactions where visual and non-verbal body language are occluded. Finally, data interpretation is subjective and influenced by the epistemic position of the research team; VH is a PhD student with a physiotherapy background where the character of patient's pain is important for ascertaining the involved structure. Consequently, certain patterns in the patient's history may have been more apparent than would have been if approached through a different lens. To ensure confirmability, data interpretation was regularly discussed and revised in collaboration with senior academic GPs (FMW, JU-S) and PPI representatives (MJ and JL).

### Comparison with existing research

As previously mentioned, this is the only study to directly examine how patients describe abdominal sensations with what GPs subsequently document in the context of early cancer diagnosis.

Existing literature has shown that patients with cancer describe abdominal sensations using nuanced vocabulary that extends beyond well-defined medical nomenclature, which they sometimes conflate. Humphrys *et al* found patients with oesophago-gastric cancer used terms such as 'stringy goo' to describe their phlegm and articulated that their legs were 'getting thinner' to convey weight loss.[17] In the same study, patients held overlapping conceptualisations of heartburn, dyspepsia and indigestion which were inconsistent with the medical definition of each symptom.[17] For example, patients described heartburn, which is defined by the American Gastroenterological Association as a 'burning sensation in the retrosternal area', in terms of acid reflux or dyspepsia.[17 41] Bankhead *et al* similarly noted interchangeable use of medical terms in interviews with women with suspected ovarian cancer, where increased abdominal size was referred to as 'bloating' instead of the appropriate medical term 'abdominal distension'.[18] The most comparable study available is a recently published prospective cohort study of over 300 patients, which examined agreement of GP and patient-reported abdominal symptoms at referral for faecal immunochemical testing for suspected colorectal cancer.[19] Considerable variation in agreement was observed by symptom type and symptom severity: GPs most frequently reported abdominal pain and diarrhoea compared with reflux and constipation and increasingly reported abdominal pain and diarrhoea as patient-reported symptom severity on a six-point Likert scale increased (eg, 63.6% and 53.9%, respectively, for minor discomfort vs 83.3% and 94.4%, respectively, for very severe discomfort). However, because studies have typically used interviews[17 18] and survey methods[19] for patients referred for suspected cancer or recently diagnosed with cancer, insights yielded may not fully capture how patients communicate bodily sensations to GPs that are yet to prompt suspicion. As we explored how patients communicate abdominal sensations during primary care consultations before diagnosis, our findings extend existing literature by identifying nuances in how patients may convey bodily sensations during earlier consultations. Further, our findings suggest that discrepancies in GPs reporting of abdominal sensations may continue into the medical records depending how they are communicated.

### Future research

The finding that well-defined or painful abdominal sensations may be more salient to GPs than vague bodily sensations expressed as 'discomfort' warrants further evaluation. Differences between the information patients communicate about their abdominal sensations and what GPs document could be clinically important given the majority of patients with cancer are diagnosed following symptomatic presentation to primary care,[42] and post hoc case reviews suggest some symptoms are interpreted incorrectly or not acted upon.[43 44] Establishing the importance of discrepancies in medical documentation requires an understanding of the meaning behind the words and phrases patients use, and the extent to which 'lay' descriptors can be reconciled with medical nomenclature. Exploring this will be vital for developing strategies that facilitate medical documentation that more closely reflects the patient's history.

The influence of identified variations in patients' descriptions of abdominal sensations on GPs subsequent use of tests and referral pathways leading to an UGI cancer diagnosis will also be important to establish. In particular, the impact on the detection of harder-to-diagnose cancers via new rapid diagnostic centres (RDCs) for concerning non-specific symptoms not qualifying for NICE 'fast-track' referral requires consideration.[45] Earlier cancer diagnosis through RDCs is contingent on prompt recognition of non-specific symptoms needing investigation, which may be hindered if potentially concerning vague symptoms or 'discomfort' are not recognised as important. Gender differences in the way patients describe symptoms of angina has been reported, with male patients using succinct language compared with female patients who used a wider repertoire of descriptors.[46] Thus, variations in descriptions of abdominal sensations by patient gender and other characteristics such as socioeconomic

status should be explored given the potential implications for inequalities in the timely investigation of cancer. We identify a range of symptom features (eg, aggravating and relieving factors and duration) capturing the nature and quality of patients' bodily sensations which are yet to be evaluated in studies developing risk prediction models for early cancer detection that limit inclusion to symptom type.[20 21 47 48] While most patients with cancer are diagnosed after a referral from their GP, other primary care practitioners are also involved in the timely recognition of symptoms of possible cancer warranting investigation.[49] As patients may feel more at ease communicating with such practitioners,[50] future similar studies that include a broader range of clinicians will also be important.

### Implications for policy and practice

Abdominal sensations described in well-defined terms or as 'pain' may be deemed more significant by GPs than those communicated as 'discomfort' or described more vaguely, even though bodily sensations by are subjectively experienced states.[14] This could mean that patients more able to present their abdominal sensations succinctly or in terminology akin to medical terminology may be perceived to have more concerning symptoms which are investigated more promptly than patients with serious illness who do not articulate their symptoms as definitively. The integrity of current and future risk prediction models for UGI cancer (and other cancers of the abdomen),[20 21 51 52] which harvest coded and/or free-text symptoms from medical records may be undermined if consultation notes do not reflect the patient's history. Similarly, growing use of 'chatbot' or App technologies, which use symptoms entered remotely by patients to suggest clinical management may be flawed if information about abdominal symptoms relies solely on medical terminology.[53]

GPs need to be cognizant that patients describe abdominal sensations in a variety of ways that differ from medical terminology which may not be indicative of their significance. Policy makers and researchers should avoid interventions or guidance that reinforce false dichotomies that give precedence to abdominal symptoms packaged in medical language over 'lay' expressions or 'discomfort'. Instead, strategies that support GPs to interpret and act on patient meaning rather than clinically exacting terminology across a broader range of symptom features is needed.

**Author affiliations**
[1]Department of Public Health and Primary Care, University of Cambridge, Cambridge, UK
[2]Nuffield Department of Primary Care Health Sciences, University of Oxford, Oxford, UK
[3]Department of Family Medicine, University of Washington, Seattle, Washington, USA
[4]Centre for Cancer Research and General Practice and Primary Care Academic Centre, University of Melbourne Victorian Comprehensive Cancer Centre, Parkville, Victoria, Australia
[5]Houston Veterans Affairs Center for Innovations in Quality, Effectiveness and Safety, Michael E DeBakey VA Medical Center, Houston, Texas, USA
[6]Department of Medicine, Baylor College of Medicine, Houston, Texas, USA
[7]Wolfson Institute of Population Health, Barts and the London School of Medicine and Dentistry, Queen Mary University of London, London, UK

**Contributors** VH and FMW designed the study. RB collected the primary data upon which this analysis was conducted. VH led the analysis and interpretation of the data with support from JU-S, SA and FMW. VH drafted the manuscript. JU-S, SA, FMW, RB, JL, MJ, MT, JE, HS and FMW critically reviewed and commented on the manuscript. FMW is the guarantor for this work. All authors approved the final manuscript for submission.

**Funding** This work is supported by the CanTest Collaborative which is funded by Cancer Research UK (C8640/A23385), where VH is a PhD student, FMW is Director, and MT, HS and JE are Associate Directors. SA is supported by Cancer Research UK grants (C12292/A20861). The Bristol Archive Project was funded by the National Institute for Health Research (NIHR) School for Primary Care Research (208) and the South West GP Trust.

**Competing interests** None declared.

**Patient and public involvement** Patients and/or the public were involved in the design, or conduct, or reporting, or dissemination plans of this research. Refer to the Methods section for further details.

**Patient consent for publication** Not applicable.

**Ethics approval** Research ethics approval was obtained from the East of Scotland Research Ethics Committee (REC) prior to commencing the study (reference: 19/ES/0057).

**Provenance and peer review** Not commissioned; externally peer reviewed.

**Data availability statement** No data are available.

**ORCID iDs**
Victoria Hardy http://orcid.org/0000-0002-8620-4404
Juliet Usher-Smith http://orcid.org/0000-0002-8501-2531
Stephanie Archer http://orcid.org/0000-0003-1349-7178
Jon Emery http://orcid.org/0000-0002-5274-6336
Hardeep Singh http://orcid.org/0000-0002-4419-8974
Fiona M Walter http://orcid.org/0000-0002-7191-6476

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
