## [Reviewer comments · BMJ Open]

ARTICLE DETAILS

TITLE (PROVISIONAL)	Agreement between patient's description of abdominal symptoms of possible upper gastrointestinal cancer and General Practitioner consultation notes: a qualitative analysis of video-recorded UK primary care consultation data
AUTHORS	Hardy, Victoria; Usher-Smith, Juliet; Archer, Stephanie; Barnes, Rebecca; Lancaster, John; Johnson, Margaret; Thompson, Matthew; Emery, Jon; Singh, Hardeep; Walter, Fiona

VERSION 1 – REVIEW

REVIEWER	Black, Georgia University College London, Applied Health Research
REVIEW RETURNED	19-Nov-2021

GENERAL COMMENTS	Thank you for the opportunity to review this interesting and engaging paper. My overall impression is one of a very thoroughly conducted and reported study, taking a novel approach to the well established problem of managing non-specific abdominal symptoms in primary care. The particularly novel aspect is to consider how symptoms are written into the patient record in contrast with verbal information given by the patient. This is a crucial part of the management of non-specific symptoms, where we know that multiple consultations are likely, and the recent medical history (and patient record) is of paramount significance. A particular strength of the paper is to include behavioural gestures in the analysis, along with the content analysis of verbal data. This tells us a lot about how patients describe their own symptoms and advocate for further investigation, for example using large hand gesticulations to supplement descriptions of pain. The use of the Cambridge-Calgary model as a broad organisation framework is effective here, and the figures support this. I cannot find fault with any aspect of the write up and discussion. I would strongly recommend publication, and I endorse the authors' views that further studies of this type would be a valuable addition to the literature. A very minor quibble is that I cannot find any mention of references 57 and 58 in the final paragraph despite their inclusion in the reference list, but it may be that I have missed them somewhere.
--

REVIEWER	Barratt, Julian Health Education England, Faculty of Advancing Practice
REVIEW RETURNED	03-Feb-2022

GENERAL COMMENTS	Thank you for preparing this interesting paper for BMJ Open which
---

	is within my area of subject expertise - consultation communication. Please may I note some minor points of clarification?  1. On page 4 and into page 5 - it is refers to GPs' documentation of patients' bodily sensations and GPs' estimates of cancer risk - but we're now working in a world of multi-professional general practice, where other types of clinicians working in advanced practice roles, such as advanced nurse practitioners or advanced clinical practitioners and the suchlike are assessing, diagnosing, and managing patients on the same basis as their general practitioner colleagues. My question is, should the evident reality of the multi-professional dimension of general practice be reflected in more inclusive terminology such as 'general practice clinicians'? 2. I was wondering why and how a convenience sample was selected since no new collection took place for the study as it was working with an existing archive of recorded general practice consultations? Please can this reasoning be elaborated further? 3. I was wondering what specific type of 'qualitative content analysis' was used the data analysis and why it was used? Isn't saying 'qualitative content analysis' was used for data analysis, but with no further specification noted, similar to saying 'statistical analysis' was used for data analysis without reporting what any of the tests used were? 4. The usage of the Calgary-Cambridge Guide is described, but I was wondering why it was used? Also, was consideration given to using any alternatives such as interaction analysis systems, for example, RIAS? 5. The sections on pages 15-17 are written very much as if the findings of the study only have meaning and relevance to GPs, but I think they have relevance and meaning for the whole multi-professional team who work alongside GPs at that similarly advanced level of practice, such as advanced nurse practitioners or advanced clinical practitioners. Can the sections on pages 15-17 be made more inclusive of the contemporary multi-professional composition of advanced practice clinical teams in general practice?
--	--

VERSION 1 – AUTHOR RESPONSE

Reviewer: 1

Dr Georgia Black, University College London Comments to the Author:

Thank you for the opportunity to review this interesting and engaging paper. My overall impression is one of a very thoroughly conducted and reported study, taking a novel approach to the well-established problem of managing non-specific abdominal symptoms in primary care. The particularly novel aspect is to consider how symptoms are written into the patient record in contrast with verbal information given by the patient. This is a crucial part of the management of non-specific symptoms, where we know that multiple consultations are likely, and the recent medical history (and patient record) is of paramount significance.

A particular strength of the paper is to include behavioural gestures in the analysis, along with the content analysis of verbal data. This tells us a lot about how patients describe their own symptoms and advocate for further investigation, for example using large hand gesticulations to supplement

descriptions of pain. The use of the Cambridge-Calgary model as a broad organisation framework is effective here, and the figures support this.

I cannot find fault with any aspect of the write up and discussion. I would strongly recommend publication, and I endorse the authors' views that further studies of this type would be a valuable addition to the literature. A very minor quibble is that I cannot find any mention of references 57 and 58 in the final paragraph despite their inclusion in the reference list, but it may be that I have missed them somewhere.

Thank you so much for this feedback and for highlighting the two references which were indeed missing from the manuscript. We have updated the manuscript to address recommendations raised by Reviewer 2 and to further enhance readership. Citations for all references are now included in the body of the manuscript.

Upon reflection, we also considered it important to highlight that this was an analysis conducted on data collected before the Covid pandemic, when consulting patterns were exclusively face-to-face. As a result, our findings may not be transferable to teleconsultations. We have therefore included the following text under 'Strengths and limitations' on page 16:

“Although face-to-face consultations are gradually resuming, remote consultations remain ubiquitous and may affect how patients communicate and what GPs document. Therefore, our findings are unlikely to reflect communication or documentation practices occurring from remote patient-GP interactions where visual and non-verbal body language are occluded.”

Reviewer: 2

Dr Julian Barratt, Health Education England Comments to the Author:

Thank you for preparing this interesting paper for BMJ Open which is within my area of subject expertise - consultation communication. Please may I note some minor points of clarification?

1. On page 4 and into page 5 - it refers to GPs' documentation of patients' bodily sensations and GPs' estimates of cancer risk - but we're now working in a world of multi-professional general practice, where other types of clinicians working in advanced practice roles, such as advanced nurse practitioners or advanced clinical practitioners and the suchlike are assessing, diagnosing, and managing patients on the same basis as their general practitioner colleagues. My question is, should the evident reality of the multi-professional dimension of general practice be reflected in more inclusive terminology such as 'general practice clinicians'?

We focus on GPs (or international equivalent practitioners) in this study as currently in the UK most patients are diagnosed with cancer after a GP referral. However, we agree that in jurisdictions outside of the UK, and potentially in the UK in the future, other providers such as Physician Assistants and paramedics play an important role in the early recognition of cancer, which we acknowledge in the Introduction on page 4 as follows:

“As well as being important at the patient-level for subsequent management decisions, the ways GPs (and other equivalent primary care providers in jurisdictions outside the UK) document patients' bodily sensations in medical records is important for population approaches to early cancer detection.”

This study was conducted on data collected from English primary care consultations with GPs. Therefore, our findings have most relevance to GPs in the UK or in jurisdictions with a similar health care structure, which is why the narrative in this study is focused on GPs. We have acknowledged this as a potential limitation to the study in the Strengths and Limitations section of the Discussion on page 16 with the following text:

“The focus of this study was limited to GPs. Therefore, our findings have most applicability to GPs in the UK or countries with similar health care structure. However, we anticipate our findings will also be relevant to other primary care providers, such as physician assistance and advanced nursing practitioners seeing patients with new symptoms.”

2. I was wondering why and how a convenience sample was selected since no new collection took place for the study as it was working with an existing archive of recorded general practice consultations? Please can this reasoning be elaborated further?

We recognize this our use of the wording ‘convenience sampling’ may be confusing to the reader. We amended the text on how we sampled patients on page 6 under the heading ‘sampling’, as follows:

“As this study was an analysis of a pre-existing archived dataset and no further data collection was possible, we included all consultations of symptomatic patients that met study eligibility criteria”.

3. I was wondering what specific type of 'qualitative content analysis' was used the data analysis and why it was used? Isn't saying 'qualitative content analysis' was used for data analysis, but with no further specification noted, similar to saying 'statistical analysis' was used for data analysis without reporting what any of the tests used were?

Qualitative content analysis is a distinct qualitative analytical technique that is used to look for and textually summarize patterns within communication artifacts.(1) We specify that we undertook a ‘manifest’ qualitative content analysis to orientate the reader to the fact that we were interested in identifying explicit or more surface-level patterns in patients’ descriptions of their abdominal sensations rather than uncovering the hidden meaning, as in the case of a ‘latent’ qualitative content analysis. For reproducibility and transparency, we cite a systematic approach to the conduct of a manifest content analysis which we also describe in detail throughout the ‘Data Analysis’ section on page 6 of the manuscript. The first paragraph of the data analysis section to which your query primarily relates, has been amended to read as follows:

“Data were analysed using a manifest qualitative content analysis, following the four phases outlined by Erlingsson et al.: ‘familiarising oneself with the data’, ‘dividing and condensing the text into meaning units’, ‘formulating codes’, and ‘developing categories and themes’. A manifest qualitative content analysis was selected as it is the most suitable analytical technique for exploring and describing visible and explicit patterns (as opposed to uncovering the underlying meaning of text, characteristic of latent content analysis) across data sources that incorporate visual and textual communication data.”

4. The usage of the Calgary-Cambridge Guide is described, but I was wondering why it was used? Also, was consideration given to using any alternatives such as interaction analysis systems, for example, RIAS?

We decided to use the Calgary-Cambridge Guide over other instruments because it is an established teaching and assessment tool that is integrated into the medical education curriculum and training programmes internationally. As the objective of this study was to describe patterns in communication of abdominal symptoms, the Calgary-Cambridge Guide offered an internally valid framework containing the specific symptom features that should be explored with patients during the medical interview. We anticipated this would allow us to get closer to the text during analysis, whilst also highlighting the symptom features from the patient history GPs capture, which could inform subsequent communication training programs. The justification for using this guide has been explained in the ‘Data Analysis’ section on pages 6 and 7, and now reads as follows:

“We selected the Calgary-Cambridge Guide over other possible instruments (e.g., the Roter Interaction Analysis System) as it is an evidence-based teaching and assessment resource that is used in medical training programmes internationally to develop doctors’ verbal and non-verbal communication skills and structure consultations. The guide delineates consultations into distinct phases, including ‘initiating the session’, ‘gathering information’, ‘physical examination’, ‘explanation and planning’, and ‘closing the session’). The ‘gathering information’ phase comprises three domains: the biomedical perspective, patient perspective, and background information. Each domain specifies a range of symptom features doctors should elicit during the patient history, thereby offering an ecologically valid framework for guiding the deconstruction and analysis of patients’ descriptions and GPs’ documentation of abdominal sensations.”

5. The sections on pages 15-17 are written very much as if the findings of the study only have meaning and relevance to GPs, but I think they have relevance and meaning for the whole multi-professional team who work alongside GPs at that similarly advanced level of practice, such as advanced nurse practitioners or advanced clinical practitioners. Can the sections on pages 15-17 be made more inclusive of the contemporary multi-professional composition of advanced practice clinical teams in general practice?

As previously mentioned, at present most cancer patients are diagnosed following a GP referral. However, you make a prescient point. We agree that our findings have particular relevance for nurse practitioners in primary care settings who are currently involved in cancer screening programmes, and are likely to play an important role in early symptomatic detection of cancer in future.(2) We now acknowledge this on page 18 under ‘Future Research’ with the following text:

“While most patients with cancer are diagnosed after a referral from their GP, the expanding role of nurse practitioners in primary care settings means they are increasingly positioned to encounter patients with concerning symptoms needing investigation. As patients may feel more at ease communicating with nurses than GPs, examining the content of patients’ dialogue with nurse practitioners might offer additional valuable insight into how to unpack the patient history.”

As indicated in our response to Reviewer 1, we considered it important to highlight that this was an analysis conducted on data collected before the Covid pandemic, when consulting patterns were exclusively face-to-face. As a result, our findings may not be transferable to teleconsultations. We therefore added the following text under ‘Strengths and limitations’ on page 16:

“Although face-to-face consultations are gradually resuming, remote consultations remain ubiquitous and may affect how patients communicate and what GPs document. Therefore, our findings are unlikely to reflect communication or documentation practices occurring from remote patient-GP interactions where visual and non-verbal body language are occluded.”

We have also re-ordered the arguments in the discussion to improve readership.

References

1. Erlingsson C, Brysiewicz P. A hands-on guide to doing content analysis. *African J Emerg Med* [Internet]. 2017;7(3):93–9. Available from: <http://dx.doi.org/10.1016/j.afjem.2017.08.001>
2. Skrobanski H, Ream E, Poole K, Whitaker KL. Understanding primary care nurses’ contribution to cancer early diagnosis: A systematic review. *Eur J Oncol Nurs* [Internet]. 2019;41(June):149–64. Available from: <https://doi.org/10.1016/j.ejon.2019.06.007>

VERSION 2 – REVIEW

REVIEWER	Barratt, Julian Health Education England, Faculty of Advancing Practice
REVIEW RETURNED	24-Apr-2022

GENERAL COMMENTS	Thank you for preparing this revised and addressing the reviewers' comments. Thank you for clarifying the type of qualitative content analysis that is being used (manifest). Thank you also for clarifying the sampling strategy. Thank you for clarifying the selection of Calgary-Cambridge. My one remaining query lies in relation to the perceived status and activities of advanced practitioners such as advanced nurse practitioners and advanced clinical practitioners in general practice. I am very experienced in this area, having worked for a number of years as a nurse practitioner in general practice, and having taught many advanced practice trainees working in general practice, undertaken research of general practice advanced nurse practitioners' consultations, and now leading on workforce transformation in an advanced practice context. From all of that experience of advanced practice I can confidently say that in the UK clinicians working in advanced practice roles in general practice are most definitely involved in their everyday clinical work in the early recognition of cancer on a similar basis to their general practitioner colleagues. To say such work undertaken by advanced practitioners is not currently happening within the UK is not an accurate reflection of the contemporary reality of general practice. Please note I am referring here specifically to advanced practitioners from multi-professional backgrounds (for example, nurses, paramedics, pharmacists), such as advanced nurse practitioners and advanced clinical practitioners, who will have been prepared to Master's level via a MSc Advanced Clinical Practice programme. Across the country advanced practitioners are working in general practice, running sessions on much the same basis as their general practitioner colleagues with similar levels of responsibility and scope of practice, and similar ranges of patient presentations being assessed and managed. The amendments refer to 'advanced nursing practitioners' - please note this is not a consensus term - the commonly used terminology is 'advanced nurse practitioner'. Page 18 refers to 'nurse practitioners' - I would suggest amending that to 'advanced nurse practitioners'. The response comments / amendments also refer to 'Physician Assistants' and 'physician assistance'. Please note in the UK the term 'Physician Associate' is used to describe one of the groups that make up the 'medical associate professions' (MAPs) who, working at the enhanced level of practice (the step before advanced practice), support doctors in the diagnosis and management of patients. The practice of Physician Associate colleagues should not be conflated with that of advanced practitioners, as advanced practitioners work autonomously at an advanced practice level in partnership with medical colleagues, rather than working at an enhanced practice level under the supervision of a doctor, as Physician Associates do. I do think this paper needs still needs to be more inclusive of the contemporary multi-professional composition of general practice by being more cognisant of the contribution of advanced practitioners
--

	working alongside their general practitioner colleagues in very similar ways. This inclusivity needs to be reflected in context-setting section of the paper on page 4, and the subsequent sections in the latter part of the paper. This paper has some really interesting and relevant findings - but those findings are not of exclusive relevance to general practitioners.
--	--

VERSION 2 – AUTHOR RESPONSE

Reviewer: 2

Dr. Julian Barratt, Health Education England

Comments to the Author:

Thank you for preparing this revised and addressing the reviewers' comments. Thank you for clarifying the type of qualitative content analysis that is being used (manifest). Thank you also for clarifying the sampling strategy. Thank you for clarifying the selection of Calgary-Cambridge.

My one remaining query lies in relation to the perceived status and activities of advanced practitioners such as advanced nurse practitioners and advanced clinical practitioners in general practice. I am very experienced in this area, having worked for a number of years as a nurse practitioner in general practice, and having taught many advanced practice trainees working in general practice, undertaken research of general practice advanced nurse practitioners' consultations, and now leading on workforce transformation in an advanced practice context. From all of that experience of advanced practice I can confidently say that in the UK clinicians working in advanced practice roles in general practice are most definitely involved in their everyday clinical work in the early recognition of cancer on a similar basis to their general practitioner colleagues. To say such work undertaken by advanced practitioners is not currently happening within the UK is not an accurate reflection of the contemporary reality of general practice. Please note I am referring here specifically to advanced practitioners from multi-professional backgrounds (for example, nurses, paramedics, pharmacists), such as advanced nurse practitioners and advanced clinical practitioners, who will have been prepared to Master's level via a MSc Advanced Clinical Practice programme. Across the country advanced practitioners are working in general practice, running sessions on much the same basis as their general practitioner colleagues with similar levels of responsibility and scope of practice, and similar ranges of patient presentations being assessed and managed.

The amendments refer to 'advanced nursing practitioners' - please note this is not a consensus term - the commonly used terminology is 'advanced nurse practitioner'. Page 18 refers to 'nurse practitioners' - I would suggest amending that to 'advanced nurse practitioners'.

The response comments / amendments also refer to 'Physician Assistants' and 'physician assistance'. Please note in the UK the term 'Physician Associate' is used to describe one of the groups that make up the 'medical associate professions' (MAPs) who, working at the enhanced level of practice (the step before advanced practice), support doctors in the diagnosis and management of patients. The practice of Physician Associate colleagues should not be conflated with that of advanced practitioners, as advanced practitioners work autonomously at an advanced practice level in partnership with medical colleagues, rather than working at an enhanced practice level under the supervision of a doctor, as Physician Associates do.

I do think this paper still needs to be more inclusive of the contemporary multi-professional composition of general practice by being more cognisant of the contribution of advanced practitioners working alongside their general practitioner colleagues in very similar ways. This inclusivity needs to be reflected in context-setting section of the paper on page 4, and the subsequent sections in the latter part of the paper. This paper has some really interesting and relevant findings - but those findings are not of exclusive relevance to general practitioners.

Thank you for clarifying the accepted British terminology for advanced clinical practitioners. As recommended, we now refer to 'advanced nurse practitioners' and 'physician associates'. Please revert to pages 16 and 18 of the manuscript.

This study used data from consultations with GPs to be comparable and extend existing primary care literature that has primarily focused on GPs appraisal and documentation of patients symptoms. However we do agree that other primary care practitioners are involved in the recognition and management of possible cancer symptoms and the findings from this study will be of relevance to them. To reflect that we have made the following changes to the introduction section:

“Although other healthcare practitioners in primary care play an important role in the early recognition of symptoms of possible cancer, this study builds on existing observational research that has examined GP’s appraisal and documentation of patient’s symptoms”

And also to the discussion section:

“Our findings may also be relevant to other primary care providers in the UK. For example, advanced nurse practitioners and physician associates who also see and manage patients with symptoms that may be caused by cancer.” [strengths and limitations, page 16]

We have also added the following text recommending that, because of the important role such practitioners play in the recognition of cancer symptoms, future research in this area may benefit from taking a more multidisciplinary approach:

“Future similar studies examining the fidelity of medical documentation among a broader range of clinicians will also be important.” [future research, page 18]

VERSION 3 – REVIEW

REVIEWER	Barratt, Julian Health Education England, Faculty of Advancing Practice
REVIEW RETURNED	22-Sep-2022
GENERAL COMMENTS	Thanks for preparing this revised version of the paper, which addressed the comments related to inclusion of advanced practitioners in relation to the wider relevance of the of study beyond GP colleagues. Please may I make one minor point of clarification for consideration? Page 5: "Although other healthcare practitioners in primary care play an important role in the early recognition of symptoms of possible cancer" would be more accurately phrased as "Although other healthcare practitioners working at an advanced level of practice in primary care play an important role in the early recognition of symptoms of possible cancer".

VERSION 3 – AUTHOR RESPONSE

Reviewer: 2

Dr. Julian Barratt, Health Education England

Comments to the Author:

Thanks for preparing this revised version of the paper, which addressed the comments related to inclusion of advanced practitioners in relation to the wider relevance of the of study beyond GP colleagues.

Please may I make one minor point of clarification for consideration?

Page 5: "Although other healthcare practitioners in primary care play an important role in the early recognition of symptoms of possible cancer" would be more accurately phrased as "Although other healthcare practitioners working at an advanced level of practice in primary care play an important role in the early recognition of symptoms of possible cancer".

The sentence has been revised as requested to: "Although other healthcare practitioners working at an advanced level of practice in primary care play an important role in the early recognition of symptoms of possible cancer...."